# A film-based intervention to reduce child maltreatment among migrant and displaced families from Myanmar: Protocol of a pragmatic cluster randomized controlled trial

Amanda Sim[1]*, Tawanchai Jirapramukpitak[2], Stephanie Eagling-Peche[3⊛], Khaing Zar Lwin[2⊛], G. J. Melendez-Torres[4], Andrea Gonzalez[1], Nway Nway Oo[5], Ivet Castello Mitjans[6], Mary Soan[7], Sureeporn Punpuing[2], Catherine Lee[8], Pattraporn Chuenglertsiri[2], Th'Blay Moo[9], Eve Puffer[10]

1 Department of Psychiatry & Behavioural Neurosciences, McMaster University, Hamilton, Ontario, Canada, 2 Institute for Population and Social Research, Mahidol University, Bangkok, Thailand, 3 Department of Social Policy & Intervention, University of Oxford, Oxford, United Kingdom, 4 College of Medicine and Health, University of Exeter, Exeter, United Kingdom, 5 Mae Tao Clinic, Mae Sot, Tak, Thailand, 6 Help without Frontiers, Mae Sot, Tak, Thailand, 7 Sermpanya Foundation, Mae Sot, Tak, Thailand, 8 Department of Mental Health, Johns Hopkins Bloomberg School of Public Health, Johns Hopkins University, Baltimore, MD, United States of America, 9 Inclusive Education Foundation, Mae Sot, Tak, Thailand, 10 Department of Psychology & Neuroscience, Duke Global Health Institute, Duke University, Durham, NC, United States of America

⊛ These authors contributed equally to this work.
* siml3@mcmaster.ca

**Data Availability Statement:** No datasets were generated or analysed during the current study. All

## Abstract

### Background

Child maltreatment is a global public health crisis with negative consequences for physical and mental health. Children in low- and middle-income countries (LMIC)–particularly those affected by poverty, armed conflict, and forced migration–may be at increased risk of maltreatment due to heightened parental distress and disruptions to social support networks. Parenting interventions have been shown to reduce the risk of child maltreatment as well as improve a range of caregiver and child outcomes, yet large-scale implementation remains limited in low-resource displacement settings. This study will examine the impact of an entertainment-education narrative film intervention on reducing physical and emotional abuse and increasing positive parenting among migrant and displaced families from Myanmar living in Thailand.

### Method

The study is a pragmatic, superiority cluster randomized controlled trial with approximately 40 communities randomized to the intervention or treatment as usual arms in a 1:1 ratio. Participating families in the intervention arm will be invited to attend a community screening of the film intervention and a post-screening discussion, as well as receive a poster depicting key messages from the film. Primary outcomes are changes in physical abuse, emotional abuse, and positive parenting behaviour. Secondary outcomes include caregiver knowledge

relevant data from this study will be made available upon study completion.

**Funding:** This study is part of the Global Parenting Initiative, which is funded by The LEGO Foundation, Oak Foundation, the World Childhood Foundation, The Human Safety Net, and the UK Research and Innovation Global Challenges Research Fund (ES/S008101/1). The funders did not and will not have a role in study design, data collection and analysis, decision to publish, or preparation of the manuscript.

**Competing interests:** The authors have declared that no competing interests exist.

of positive parenting, caregiver attitudes towards harsh punishment, caregiver psychological distress, and family functioning. Outcomes will be assessed at 3 time points: baseline, 4 weeks post-intervention, and 4-month follow up. A mixed methods process evaluation will be embedded within the trial to assess intervention delivery, acceptability, perceived impacts, and potential mechanisms of change.

## Discussion

To our knowledge, this study will be the first randomized controlled trial evaluation of a film-based intervention to reduce child maltreatment among migrant and displaced families in a LMIC. An integrated knowledge translation approach will inform uptake of study findings and application to potential scale up pending evaluation results.

## Trial registration

The study was prospectively registered with the Thai Clinical Trials Registry on 22 February 2023 (TCTR20230222005).

## Introduction

### Background and rationale

Child maltreatment–defined as physical, sexual, and emotional abuse or neglect–is a global public health crisis with self-reported prevalence rates worldwide ranging from 22.6% for physical abuse to 36.3% for emotional abuse [1]. Most child maltreatment is perpetrated in the home by parents or parental guardians [2, 3]. The potentially lifelong and intergenerational impacts of child maltreatment are well-documented and include mental health disorders (e.g., depression, anxiety, childhood behavioural problems), alcohol and drug abuse, suicidal behaviour, risky sexual behaviour, obesity, and increased risk of future violence perpetration and victimization [4, 5]. While child maltreatment is a global problem prevalent in both high-income and low- and middle-income countries (LMICs), evidence suggests that children in LMICs–particularly those living at the intersections of poverty, armed conflict and forced migration–may be at increased risk of victimization [3, 6–8]. War and war-induced displacement have been shown to increase known risk factors for child maltreatment including household poverty, parental psychological distress, and family dysfunction [3, 9, 10]. Growing evidence from low-resource, forced migration contexts suggests that exposure to war and displacement-related stressors heighten parental distress, weaken coping and emotion regulation strategies, and interact with cultural norms to increase violence against children in the home [11, 12]. Disruptions to social services and social support networks due to war and displacement further degrade caregivers' capacity to protect children and provide positive, nurturing care in the face of widespread violence and instability [9].

Interventions that support parents and caregivers to practice positive parenting skills and behaviours have been shown to reduce the risk of child maltreatment as well as improve a range of caregiver and child mental health and behavioural outcomes [13, 14]. While parenting interventions can be delivered through different modalities (e.g., individual/family-based, group-based, virtual, in-person), they typically consist of a structured series of sessions built around evidence-based components for increasing positive parent-child interaction and reducing harsh parenting. In view of robust evidence of their effectiveness, parenting

interventions are included as one of the seven INSPIRE strategies advocated by the World Health Organization for ending violence against children [15]. However, large-scale implementation of parenting interventions in LMICs remains limited, particularly in humanitarian and displacement settings where the need is arguably the greatest [16]. Delivery of parenting support to migrant and displaced families in LMICs is constrained by structural and contextual factors including limited infrastructure and workforce, inadequate financial resources, and low levels of education and basic and digital literacy among the target population [17]. Reach may be further compromised by ambivalence on the part of host governments to fund services for refugees and migrants, resulting in fragmented and under-resourced delivery of parenting support to these populations.

Film-based interventions, a form of entertainment-education, has the potential to overcome many of these barriers and reach large audiences at relatively low cost with minimal infrastructure and human resource support [18]. While there is accumulating evidence that film and other narrative entertainment-education approaches can be effective at addressing certain health outcomes in LMICs including HIV prevention, sexual and reproductive health, and child survival, to our knowledge such approaches have not been applied to reducing child maltreatment among migrant and displaced populations in low-resource settings [19–21].

## Film-based interventions

Entertainment-education refers to "the intentional placement of educational content in entertainment messages" [22 p117]. Narrative approaches to entertainment-education, including film, are hypothesized to impact knowledge, attitudinal, and behavioural outcomes through a range of mechanisms [23]. For example, social learning theory posits that film or show characters may serve as potential role models and improve audiences' self-efficacy for adopting new behaviours [24], while constructs such as transportation (i.e., engagement in the narrative) [25] and identification (i.e., involvement with characters) [26] highlight the potential for narrative approaches to reduce counter-arguing and resistance to intervention messages. By activating both emotional and cognitive processes, narrative film-based interventions may be able to influence culturally entrenched values and practices that contribute to negative outcomes for children and families.

There is currently limited evidence of the potential for film-based interventions to reduce child maltreatment or other forms of violence in the home, particularly in LMICs where most children and adolescents reside. A systematic review of universal interventions with a media component aimed at preventing child physical abuse found only 17 studies in five countries, all high-income [27]. While quality of studies was generally low, the review found significant reductions in child abuse incidence as well as a range of related outcomes including dysfunctional parenting. Only a few studies have examined entertainment-education interventions to address family violence in LMICs [28, 29] and these have focused on violence against women. For instance, evaluation of *MTV Shuga* in Nigeria found that exposure to this educational TV series improved men's attitudes towards gender-based violence eight months later [30]. This emerging evidence underscores the need for greater investment in developing and rigorously evaluating the use of narrative entertainment-education approaches to reduce child maltreatment, particularly in low-resource displacement settings where access to parenting support is extremely limited.

## Objectives

The current study will evaluate the impact of a film-based intervention on reducing two forms of child maltreatment, physical and emotional abuse, and increasing positive parenting

behaviour among migrant and displaced families from Myanmar living in Thailand. To our knowledge, this will be the first randomized controlled trial evaluation of a film-based intervention to reduce child maltreatment among migrant and displaced families in a LMIC. A mixed methods process evaluation will be embedded within the trial to assess intervention delivery, acceptability, perceived impacts, and potential mechanisms of change. The study is being implemented as part of the Global Parenting Initiative, a global consortium of academic and research institutions and implementing organizations aiming to provide access to free, evidence-based, playful parenting support to promote child learning and prevent violence at scale in the Global South [31].

## Materials and methods

### Aims

The primary study aim is to evaluate the effectiveness of a film-based intervention for reducing caregiver-reported physical and emotional abuse and increasing positive parenting. Secondary aims are to assess the impacts of the intervention on caregiver knowledge of positive parenting, caregiver attitudes towards harsh punishment, caregiver psychological distress, and family functioning. Additional aims are to explore intervention impacts on caregiver coping, behaviours to support early learning, social support, and child internalizing and externalizing symptoms. The study will also examine potential treatment moderators (e.g., lifetime trauma exposure, length of displacement) and mediators (e.g., caregiver psychological distress), as well as acceptability and perceived intervention impacts and mechanisms of change.

### Trial design

The study is a pragmatic, superiority cluster randomized controlled trial with approximately 40 communities (clusters) randomized to the intervention or treatment as usual (TAU) arms in a 1:1 ratio. Participating families in communities allocated to the intervention arm will be invited to attend a community screening of the film intervention and a post-screening discussion. Each family will also receive a poster depicting key messages from the film.

Primary outcomes for the trial are changes in (1) physical and emotional abuse; and (2) positive parenting behaviour. Secondary outcomes include caregiver knowledge of positive parenting, caregiver attitudes towards harsh punishment, caregiver psychological distress, and family functioning. Exploratory outcomes include caregiver coping, behaviours to support early learning and education, social support, and child internalizing and externalizing symptoms. Outcomes will be assessed at baseline prior to randomization (T0), approximately 4 weeks post-intervention (T1), and 4-month follow up (T2) (S1 File). Qualitative group interviews with caregivers and adolescents will be conducted at T1 to examine acceptability of the film-based intervention and perceived impacts on caregiver, child, and family outcomes.

### Study setting and implementation partners

The study will be conducted in Tak province in Thailand on the border with Myanmar. There is a long-standing history of migration and displacement from Myanmar into Tak province due to decades of armed conflict and political and economic instability in Myanmar [32]. As a result of the *coup d'état* in Myanmar in February 2021, the area has seen a significant influx of individuals from Myanmar across the border into Tak province [33]. A 2022 report indicates that Thailand hosts 658,023 individuals from populations of concern including an estimated 91,401 refugees from Myanmar who live in the 9 refugee camps along the border [33]. However, this figure does not include the large number of migrants and displaced people from

Myanmar who live outside the camps. The 2019 Thailand Migration Report estimates that there are 3.9 million migrants living in Thailand, with approximately 70% originating from Myanmar [34]. Thailand is not a signatory to the 1951 Refugee Convention and there is currently no national legislation or legal framework governing the definition, rights, and protection of refugees, migrants, and other displaced people in Thailand [34].

Many migrants and displaced people from Myanmar do not have work or residence permits and are at risk of deportation, exploitative working conditions, and inequitable access to education, health, and other essential services [33, 35]. Reduced opportunities for mobility and employment as a result of the COVID-19 pandemic have further increased levels of poverty and food insecurity, which in turn has elevated risk of child maltreatment and other negative child and family outcomes [36]. Recent studies with migrant and displaced families from Myanmar in Tak Province highlight concerns about physical and emotional abuse in the home, child labour and neglect, as well as growing mental health difficulties such as depression and anxiety among both parents and children [32, 37]. Results from the 2015 Myanmar Demographic Health Survey indicate high prevalence of harsh discipline in Myanmar, with 74% of children aged 2 to 14 experiencing emotional abuse and 43% experiencing physical abuse [38]. These findings suggest that the use of harsh punishment is widespread among the Myanmar population and may be further intensified by the chronic adversity experienced by migrant and displaced families living across the border in Thailand.

The study will be implemented in partnership with local non-governmental and community-based organizations Mae Tao Clinic, Help Without Frontiers Thailand Foundation, Inclusive Education Foundation, and Sermpanya Foundation. All partner organizations have extensive experience providing health, education, and psychosocial support to migrant and displaced families in Tak province. We refer to the study population as "migrant and displaced" to recognize that individuals and families from Myanmar have varying legal status in Thailand and have left their country of origin for a combination of economic and conflict-related factors [39].

## Intervention and control conditions

**Film-based intervention: "Being Family".**   The intervention consists of a 66-minute live action narrative film created in collaboration with migrant and displaced families, local partners and stakeholders, and Sermpanya Foundation, a Thai non-governmental organization that has produced and screened educational films on the Thailand-Myanmar border since 2011. Key messages depicted in the film were adapted from Parenting for Lifelong Health (PLH), a suite of open-access parenting programs that has been tested in 11 RCTs in LMICs including Thailand [40, 41]. The film focuses on four key messages: the importance of play and positive parent-child interaction; praise and positive family communication; rules and non-violent consequences for behaviour management; and parental coping. The characters and scenarios depicted in the film reflect different ages and developmental stages to ensure that the film would be relevant to families with children across a wide age range.

Intervention development took place over a period of 12 months and was informed by principles of human-centred design [42]. We first conducted formative qualitative research with parents/caregivers and adolescents from Myanmar living in Tak province to understand existing parenting practices, risk factors for child maltreatment, and acceptability of potential parenting and coping strategies. Formative research findings were used to inform script development, which was led by a refugee filmmaker at Sermpanya Foundation in close collaboration with the research team. An advisory committee composed of community-based organizations in Tak province and a group of parents/caregivers from Myanmar participated in a

read-through of the script and provided feedback on characters, plot, and messages which was incorporated into the final version of the script. Filming on location in Tak province took place from December 2022 to April 2023. All cast and crew members were migrants and displaced individuals from Myanmar who provided ongoing feedback to ensure resonance with the social and cultural context. A final draft of the film was shown to the advisory committee, partner organizations, and community members from the film shooting location before being finalized for implementation in May 2023.

Sermpanya Foundation will conduct mobile screenings of the film in community locations that can accommodate large groups and that are accessible and safe for migrant and displaced families. Study participants and their families will be invited to attend the film screening and attendance will be tracked to assess intervention exposure. Other members of the same community will also be able to attend; however, individuals residing in other communities will not be admitted to minimize spill over. Immediately after the film screening, trained facilitators from local partner organizations will lead a structured discussion with audience members to reinforce the key messages and skills depicted in the film. The audience discussion will last approximately 45 minutes, after which the audience will be shown a 5-minute video featuring two community advocates summarizing the key messages from the film and audience discussion. Finally, each household will receive a poster depicting key messages.

**Treatment as usual (TAU).** TAU will consist of providing study participants with information about existing services related to child protection and safeguarding and related education and health services in Tak province. In collaboration with local partners, we developed a list of services with brief descriptions including information on how to access services. Caregivers in both the intervention and control groups will receive this information via handouts distributed at the time of the baseline assessment. Caregivers' participation in any other parenting-related interventions or services during the study is permitted and will be assessed at endline (T1).

## Participants

**Communities.** An initial list of potential communities in 4 districts in Tak province (Mae Sot, Pop Phra, Mae Ramat, Tha Song Yang) was generated from a database of migrant learning centres (nonformal community-based schools attended by migrant and displaced children and youth from Myanmar) available from the Migrant Educational Coordination Center [43]. The study team and implementing partners shortlisted communities estimated to have at least 70 migrant and displaced families from Myanmar with children aged 4 to 17 years, and added other communities known to local partners that were not represented on the database of migrant learning centres. The study team then conducted meetings with formal and informal leaders and stakeholders (e.g., village leaders, teachers, community volunteers) in each of the communities on the shortlist to introduce the study, obtain information about the community (e.g., geographic boundaries, estimated population, language spoken), and gain permission to conduct the study. To the extent possible, communities will be purposively selected to ensure separation by a geographical buffer (e.g., at least one community, road, or natural structure) to reduce the risk of spillover effects. Movement between communities tends to be limited as many families from Myanmar do not have legal status in Thailand and are unable to travel freely.

**Participant inclusion/exclusion criteria.** Participants are caregivers of at least one child aged 4 to 17 years and adolescents aged 12 to 17 years. Caregivers are defined as parents or guardians with self-reported primary responsibility for the care of a minor child under the age of 18 years living in the same household. There is no requirement for a biological relationship between the caregiver and child. Caregivers will be recruited if they meet the following inclusion criteria: primary caregiver of at least one child aged 4 to 17 years; from Myanmar; currently

residing in the study site; and, conversant in Burmese. In addition, adolescents aged between 12 and 17 years who reside in the same household as the caregiver will be eligible for focus group discussions at T1. Individuals will be excluded if they have significant or severe cognitive, neurological, or developmental impairments that render them unable to provide informed consent, as assessed by data collection staff. Caregivers and adolescents who reside in institutions or group homes (e.g., boarding houses, orphanages) will be excluded from the study.

## Procedures

**Recruitment, informed consent and data collection.** Formal and informal community leaders and other community members who are knowledgeable about children and families (e.g., teachers, community volunteers) will be trained on the participant inclusion and exclusion criteria and asked to recruit eligible caregivers residing in their community. Caregiver information (e.g., name, gender, phone number) will be registered either at the time of recruitment or immediately prior to the baseline survey.

Informed consent and data collection will take place in small groups of approximately 5 participants, facilitated by trained data collection staff. Each participant will be provided with an electronic tablet to enter their responses during the assessment. Staff will begin by reading out the study information sheet to caregivers and provide opportunities for questions. If individuals agree to take part, they will check the appropriate box on the informed consent statement displayed on their electronic tablet to indicate consent. Participants will also be provided with a hard copy of the study information sheet with contact information for the local ethics board and research team. Due to low literacy levels among the study population, data collection staff will read out the survey items and response options while participants follow along and enter responses on their tablet. Pictorial Likert scales and other visual aids will also be displayed on the tablet to facilitate comprehension of response options.

Assessments will be conducted in a private place that is convenient and acceptable to participants (e.g., home, school). Caregiver and adolescent participants will receive a token of appreciation (e.g., mobile phone credit, stationery, cooking supplies) with a value of 140 THB (approximately USD $4) at each assessment point. In addition, a lottery will be conducted in each community after each round of data collection for a chance to win one of 5 prizes worth 200 THB each.

**Randomization.** Participant recruitment, baseline assessment, randomization, and intervention delivery will take place on a rolling basis in blocks of 4 communities. Blocks will be created within strata (i.e., Mae Sot district vs. all other districts) to minimize potential imbalance between study arms as implementing partners have observed differences in sociodemographic characteristics of families residing in the more urban Mae Sot district compared to other more rural districts. Following participant recruitment and baseline data collection in each of the 4 communities in the block, the trial statistician (GJMT) will perform randomization using a random number generator in Stata.

**Blinding.** The trial statistician (GJMT) conducting randomization and analyses will be blinded to condition. Blinding of participants and implementers will not be possible as they will know whether or not they are receiving the intervention. Data collection staff will assess intervention exposure at T1 and hence will not be blinded to condition. To reduce risk of bias, intervention exposure will be assessed at the end of the endline survey.

## Outcomes

Existing Burmese translations of measures were obtained from study authors where available. All other measures were translated into Burmese following recommended procedures

including forward translation to Burmese, back translation to English by an independent translator, and consultations with bilingual team members to review and resolve any discrepancies [44]. Survey instruments were pilot tested with the target population to assess acceptability, relevance, comprehensibility, and length prior to data collection. The survey included approximately 100 items and took 1–2 hours to complete, which was deemed acceptable during piloting.

**Primary outcomes.** Physical abuse and emotional abuse will be assessed separately using items from the International Society for Prevention of Child Abuse and Neglect Child Abuse Screening Tool (ICAST-Trial; physical abuse: 4 items, range 0–32; emotional abuse: 9 items, range 0–72) [45]. Positive parenting will be assessed using an adapted version of the Parent Behaviour Inventory (PBI, 10 items, range 0–30) [46]. The PBI was previously developed and used by authors AS and EP for a trial of a family skills intervention on the Thailand-Myanmar border. These measures will be administered to caregivers at all 3 time points.

**Secondary outcomes.** Caregiver knowledge of positive parenting will be assessed by asking caregivers to indicate their agreement with 5 items developed for this study (e.g., *"Praising children is an effective way to teach children how to behave well."*). Caregiver attitudes towards physical punishment will be assessed by asking caregivers to indicate their agreement with a single item from the 2015 Myanmar Demographic Health Survey (e.g., *"Do you believe that in order to bring up your child properly, you need to physically punish him/her?"*) [38], as well as 4 additional items adapted from the UNICEF Multiple Indicator Cluster Surveys [47].

Caregiver psychological distress will be assessed by the 10-item version of the Hopkins Symptom Checklist (HSCL) [48]. The HSCL has been used extensively with refugees and migrants from Myanmar [49–51]. Family functioning will be assessed with caregivers and adolescents using 6 items from the Burmese Family Functioning Scale, developed by authors AS and EP from formative qualitative research and previously used in a trial of a family skills intervention on the Thailand-Myanmar border [46]. We will also transform estimates of physical and emotional abuse into binary variables (yes/no) to investigate proportion reporting any of each type of abuse.

**Exploratory outcomes.** Caregiver coping and stress management will be assessed using 4 items drawn from the PBI (e.g., *"In the past 4 weeks until now, I felt too stressed to spend time with my child."*) [46]. Caregiver behaviours to support early learning will be assessed using 6 items drawn from the 2015 Myanmar Demographic Health Survey (e.g., *"In the past 7 days, how often did you read books to or look at picture books with your child?"*) and administered only to caregivers of index children aged 4 to 8 years [38]. For caregivers with index children currently enrolled in school, behaviours to support education will be assessed using 6 items (e.g., *"In the last 30 days, how often did you praise your child for working hard at school?"*) adapted from Ceballo et al. [52].

To assess caregiver social support, we will use an abbreviated version of the Medical Outcomes Study Social Support Scale (MOS-SSS) consisting of 5 items from the tangible support, emotional-informational support, affectionate support, and positive social interaction support subscales [53]. The MOS-SSS was previously used in a study with Burmese adolescents living on the Thailand-Myanmar border [48]. Caregiver-reported child internalizing symptoms will be assessed by the Mood and Feelings Questionnaire (MFQ) [54], while externalizing symptoms will be assessed by the externalizing subscale of the Child and Adolescent Behaviour Inventory (CABI) [55].

**Other measures.** Sociodemographic characteristics (e.g. age, education level, length of stay in Thailand), caregivers' lifetime trauma exposure, daily stressors, and adverse and positive childhood experiences will also be assessed to characterize the study sample and to examine potential moderators of treatment effects. Caregivers' lifetime trauma exposure will be

assessed at T0 using 6 items adapted from the Harvard Trauma Questionnaire [56]. Each item is scored as "yes" or "no" for occurrence regardless of when or where it occurred. Caregivers' experience of daily stressors will be assessed at T0 using 9 items adapted from the Humanitarian Emergency Settings Perceived Needs Scale (HESPER) developed by the World Health Organization [57] with the addition of contextually relevant items developed from formative qualitative research. Caregivers' adverse childhood experiences will be assessed at T2 using items adapted from the Adverse Childhood Experiences International Questionnaire (ACE-IQ) developed by the World Health Organization [58]. Caregivers' positive childhood experiences will be assessed at T2 using items from the Benevolent Child Experiences Scale, which assesses positive early life experiences in adults with histories of adversity [59].

Finally, the endline survey at T1 will include assessment of intervention exposure (including in the control arm to assess spillover), transportation (i.e., the extent to which viewers are transported into or engaged in the narrative) [25], identification with characters, and realism and relevance of the film [60]. Participants will also be asked if they attended the film screening with other family members or discussed the film with family or community members to assess diffusion.

## Sample size calculation

The sample size calculation was performed by the trial statistician GJMT and based on a two-group comparison of one of the primary outcomes (physical abuse, emotional abuse, or positive parenting) assessed at T1. Assuming a significance level of 5% with an intra-cluster correlation coefficient (ICC) of 0.02 and a two-tailed test with 0.9 power, 40 clusters with 50 caregivers per cluster would be required to detect an effect size of 0.2. The total sample size was set at 2,200 caregivers (40 clusters with 55 families per cluster) to account for up to 10% attrition. Our estimate of expected effect size was based on a review of parenting interventions for child maltreatment prevention which found an average effect of 0.2 [61]. It is possible that the number of recruited caregivers will vary in each community. We will continuously monitor realized power as we enrol caregivers into the study and stop recruitment once the harmonic mean of cluster size and number of clusters reach 90% power, or until we reach maximum capacity, whichever comes first.

## Statistical analysis

Trial results will be reported following the updated recommendations of the Consolidated Standards of Reporting Trials (CONSORT) 2010 statement: extension to cluster randomised trials [62]. Initial analyses will compare baseline characteristics of participants across the two study arms as well as participants who completed T1 assessments and those who did not.

**Caregiver analyses.** Outcomes will be analysed using three-level models with measurement wave within participant within cluster where outcomes are measured over multiple time-points. Level 1 will include a term for categorical time (T1 and T2) and the interactions between intervention and categorical time; level 2 will include terms for caregiver and child age and gender, centred at the sample mean; and level 3 will include terms for intervention and stratification. Thus, the estimate of intervention effectiveness is the interaction between intervention and categorical time. A likelihood ratio test against a model without interactions will form the test of effectiveness. The primary analysis is from an imputed and adjusted model, which will generate differences in the primary outcomes between the two study arms at T1 and T2 using 95% CIs. Crude (i.e. models including intervention, time, intervention by time interactions, and stratification) imputed and unimputed models will be used for sensitivity analysis.

Primary analyses will be conducted using the intention to treat principle. To account for dropouts in the intention-to-treat analysis, the baseline measurement will be treated as a repeated measure and estimation of the intervention effects will be via maximum likelihood. The impact of the missing data on the estimated intervention effect will be assessed by imputing missing outcome data (at least 10 imputations, using an unrestricted multilevel model with fully conditional specifications) using complete baseline and follow-up data and running the same models. Per-protocol analyses will also be conducted using only the data on participants who received the intervention as planned. All additional analyses, including analysis of potential moderators and mediators, will be detailed in a statistical analysis plan prior to unblinding. Across all analyses, two-tailed tests will be reported with a significance level of $p < 0.05$.

**Cost effectiveness analysis.** Costs will be divided into set-up costs (e.g., initial script development and film production costs) and intervention delivery costs (e.g., travel, staffing, film screening equipment and supplies). Cost data will be collected from project budget and expenditure reports and verified through consultation with project staff. Cost effectiveness analysis will include the following phases: (1) review of budget/expenditure sheets and consultation with project staff; (2) calculating costs; (3) calculating effectiveness; and (4) calculating cost-effectiveness ratios to provide the cost per standard deviation unit change in primary outcomes.

## Process evaluation

A process evaluation will be embedded in the trial, drawing on components of the RE-AIM framework to assess intervention delivery [63]. A combination of quantitative and qualitative data will be collected as part of the process evaluation. Quantitative data will include total attendance at community film screenings (including study participants and other community members) to provide an indication of reach, as well as attendance by study participants specifically to assess intervention uptake. Study participants will also be asked if they attended the film screening at the endline assessment (T1). We will not collect identifying information from film screening attendees who are not enrolled in the study. To assess fidelity of the post-film audience discussion, trained research staff will observe a random sample of at least 25% of community film screenings and complete a structured observation form to score which and to what extent components of the discussion were completed.

Qualitative data will be collected from a subsample of participating caregivers and adolescents who attended the film screening to examine acceptability, relevance, and perceived impacts of the intervention. Up to 6 focus group discussions with approximately 8 caregivers and 4 focus group discussions with approximately 8 adolescents aged 14 to 17 years will be conducted after participants complete the endline assessment. We will first purposively select up to 4 communities that have received the intervention to ensure representation from urban and rural settings. Using endline survey data, we will select caregiver and adolescent participants who report having attended the film screening and recruit them for the focus group discussions. We will stratify selection by caregiver and adolescent gender and age of index child to ensure representation across these variables. Focus group discussions will be used to examine participants' experiences and reactions to the film-based intervention, perceived impacts on primary and secondary outcomes, any unanticipated outcomes, and potential causal mechanisms. In addition, we will conduct focus group discussions with community stakeholders and implementation staff to examine feasibility, acceptability, appropriateness, relevance, and sustainability of the intervention. Focus group discussions will be led by two local research staff with qualitative research experience, and data will be transcribed and translated for thematic content analysis [64].

## Data management

Assessments will be administered using Open Data Kit (ODK) on password-protected tablets and uploaded daily onto a secure encrypted server. All submissions will be cross-checked before data are permanently removed from the tablets each week. Only authorized members of the research team will have access to data on the server. Data on the ODK server is automatically backed up daily and an additional weekly back up will be made onto a password-protected OneDrive folder accessible only to authorized research team members. Any hard copies of forms will be scanned and uploaded to the server upon completion of data collection, after which they will be permanently destroyed. Audio recordings and transcription of focus group discussions will also be stored in OneDrive; once transcription is completed and verified, audio recordings will be permanently deleted from recording devices.

All data will be de-identified prior to analysis. Participant responses will be identified only by a unique study code. Information linking the unique study code to participants' identifying information such as names and contact information will be stored separately on a password-protected OneDrive folder accessible only to authorized research team members and destroyed at the conclusion of the study. Transcripts of audio recordings of focus group discussions will be anonymized by removing all personal identifying data. Audio recordings will be destroyed upon verification of the transcripts. De-identified data will be stored securely in perpetuity for research purposes only. Participants will be able to notify the study team if they want to withdraw their data until the personal linked data is destroyed. Statistical code and fully anonymized data will be made publicly available on Oxford Figshare and OSF following UK Research and Innovation (UKRI) guidelines [65].

## Trial and adverse events monitoring

A trial management committee consisting of principal investigators, co-investigators, and research coordinators will monitor the implementation of study procedures. Any deviations from the study protocol will be reported to the trial management committee and to the research ethics committees as necessary. Significant modifications to the protocol will be submitted to the research ethics committees for approval and trial protocols will be updated in the online registry.

The research team will be trained on a safety protocol developed for this study that will include information on reporting and referral procedures if a safety concern is identified. Local implementing partners have a Child Safeguarding Policy that includes a protocol for reporting suspected child maltreatment, exploitation, or any other violation. All staff will be required to adhere to this policy. Data collection staff will be trained to record and report all adverse events (e.g., extreme participant distress, child safeguarding concerns) to the local research coordinators and co-investigators (KZL, SEP) within 24 hours. A committee comprised of the co-PI (AS), local co-investigator and protection lead (NNO), and research coordinators and co-investigators (KZL, SEP) will be responsible for reviewing all reported adverse events within 48 hours and determining any necessary action, which may include referral to the local Child Protection Unit. Information is included in the informed consent form to notify participants that disclosures of harm to self or others may be reported.

To reduce risk of participant distress, data collection staff will be trained to remind participants that they can choose to skip any questions or withdraw from the study at any time with no negative consequences. All participants will be provided information about available mental health and psychosocial support services during each assessment point. No interim analyses are planned. The local research coordinators and co-investigators are responsible for ensuring timely follow up of any adverse events.

## Ethics

Ethics approval was obtained from the Institutional Review Board of the Institute for Population and Social Research (IPSR) at Mahidol University in Thailand (COA. No. 2022/11-217), as well as the ethics committees of the research team's respective institutions. The study protocol was also reviewed by the Community Ethics Advisory Board on the Thailand-Myanmar border.

## Inclusivity in global research

Additional information regarding the ethical, cultural, and scientific considerations specific to inclusivity in global research is included in S4 File.

## Trial status

Recruitment of the first study participant began on 23 February 2023. At the time of submission, baseline data collection has been completed in 33 communities and the film-based intervention has been delivered in 2 communities.

## Discussion

To our knowledge, this will be the first randomized controlled trial of a universal film-based intervention to reduce child maltreatment among migrant and displaced families in a low-resource context. Children experiencing the triple threat of poverty, war, and displacement are at increased risk of violence at the hands of their caregivers, yet delivery of parenting and psychosocial support to caregivers is currently limited by structural and contextual barriers. Results from this study will contribute to the growing evidence around narrative entertainment-education interventions by examining their effectiveness at reducing child maltreatment–a novel intervention target for entertainment-education–in a low resource humanitarian setting. Given challenges with reach and retention of parenting interventions even in high-income settings [66], the study will also provide valuable knowledge on universal strategies to promote positive parenting and prevent child maltreatment at a population level.

Study strengths include the randomized controlled trial design with a large sample size to ensure sufficient power to detect small intervention effects at a population-level. Assessment at 3 time points allows for examination of longer-term intervention and mediation effects. The use of both qualitative and quantitative methods (including standardized measures that were previously used with the study population) enables triangulation of study findings and exploration of unanticipated effects and potential mechanisms of change. Limitations include the risk of spillover or contamination which we will monitor by tracking attendance at community film screenings and assessing intervention exposure among caregivers in both intervention and control arms. There is also some risk of bias due to the inability for participants and data collection staff to be blinded as well as the non-random recruitment and selection of communities and participants. Finally, the study is only being conducted in Burmese due to resource constraints, which may inhibit participation of individuals from minority ethnic groups who do not or prefer not to use Burmese.

Study results will be disseminated to multiple audiences including research participants, implementing organizations and policy makers in Thailand and globally, and the academic community. In addition to dissemination through peer-reviewed journal articles and conference presentations, integrated knowledge translation approaches will be used to engage partners and other knowledge users in all stages of research design, implementation, and dissemination to collaboratively identify opportunities for research uptake and scale up [67].

## Supporting information

**S1 File. SPIRIT flow diagram.**
(DOCX)

**S2 File. SPIRIT checklist.**
(DOCX)

**S3 File. Approved protocol.**
(PDF)

**S4 File. Inclusivity in global research.**
(DOCX)

## Acknowledgments

We are grateful to our partners Sermpanya Foundation, Mae Tao Clinic, Help Without Frontiers, and Inclusive Education Foundation and to all communities and families for their participation in the study.

## Author Contributions

**Conceptualization:** Amanda Sim, Tawanchai Jirapramukpitak, Eve Puffer.

**Funding acquisition:** Amanda Sim, Eve Puffer.

**Methodology:** Amanda Sim, Tawanchai Jirapramukpitak, Stephanie Eagling-Peche, Khaing Zar Lwin, G. J. Melendez-Torres, Andrea Gonzalez, Nway Nway Oo, Ivet Castello Mitjans, Mary Soan, Sureeporn Punpuing, Catherine Lee, Th'Blay Moo, Eve Puffer.

**Project administration:** Nway Nway Oo, Ivet Castello Mitjans, Th'Blay Moo.

**Supervision:** Amanda Sim.

**Writing – original draft:** Amanda Sim, Stephanie Eagling-Peche, Khaing Zar Lwin, G. J. Melendez-Torres, Eve Puffer.

**Writing – review & editing:** Amanda Sim, Tawanchai Jirapramukpitak, Stephanie Eagling-Peche, Khaing Zar Lwin, G. J. Melendez-Torres, Andrea Gonzalez, Nway Nway Oo, Ivet Castello Mitjans, Mary Soan, Sureeporn Punpuing, Catherine Lee, Pattraporn Chuenglertsiri, Th'Blay Moo, Eve Puffer.

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
