## [Decision Letter · Decision Letter 0]

4 Aug 2023

PONE-D-23-16654A film-based intervention to reduce child maltreatment among migrant and displaced families from Myanmar: Protocol of a pragmatic cluster randomized controlled trialPLOS ONE

Dear Dr. Sim,

Thank you for submitting your manuscript to PLOS ONE. After careful consideration, we feel that it has merit but does not fully meet PLOS ONE’s publication criteria as it currently stands. Therefore, we invite you to submit a revised version of the manuscript that addresses the points raised during the review process.

We look forward to receiving your revised manuscript.

Kind regards,

Jianhong Zhou

Staff Editor

PLOS ONE

Reviewers' comments:

Reviewer's Responses to Questions

**Comments to the Author**

1. Does the manuscript provide a valid rationale for the proposed study, with clearly identified and justified research questions?

Reviewer #1: Yes

Reviewer #2: Partly

2. Is the protocol technically sound and planned in a manner that will lead to a meaningful outcome and allow testing the stated hypotheses?

Reviewer #1: Yes

Reviewer #2: Partly

3. Is the methodology feasible and described in sufficient detail to allow the work to be replicable?

Reviewer #1: Yes

Reviewer #2: Yes

4. Have the authors described where all data underlying the findings will be made available when the study is complete?

Reviewer #1: Yes

Reviewer #2: Yes

5. Is the manuscript presented in an intelligible fashion and written in standard English?

Reviewer #1: Yes

Reviewer #2: Yes

6. Review Comments to the Author

You may also provide optional suggestions and comments to authors that they might find helpful in planning their study.

Reviewer #1: This protocol for a randomized controlled trial addresses an important and understudied area of research—child maltreatment in migrant and displaced families, particularly in low- and middle-income countries (LMICs). It has the potential to fill a significant gap in the literature on the prevention of child maltreatment in vulnerable populations. The protocol clearly outlines the rationale, provides detailed information about the methods (including participants, measures, and procedures), and offers a precise plan for statistical analysis and discussion. The reasoning behind choosing a film-based intervention is clearly articulated, emphasizing its potential for cost-effective, large-scale implementation in resource-limited settings. The decision to use a pragmatic, superiority cluster randomized controlled trial will allow for the investigation of the intervention's effectiveness in the “real world.” I also appreciate the use of various measures (both parent reports and adolescent reports) to evaluate the primary and secondary outcomes of the intervention. Furthermore, the use of both quantitative and qualitative methods could offer valuable insights about intervention delivery, acceptability, perceived impacts, and potential mechanisms of change. Overall, I view the protocol very positively and I am looking forward to reading about the results of this important study.

I have only one minor comment:

How did the authors of the protocol ensure that the film is adequate for the needs and situations of families with children across such a wide age range, from 4 to 17 years?

Reviewer #2: The manuscript is well-written and addresses the important social issue of child maltreatment through a novel approach, and with a vulnerable population.

My questions and concerns about the manuscript are three-fold:

1) Why submit the article at this stage, before data has been collected and analyzed? While it appears to be a novel approach, the manuscript seems incomplete without an analysis of the results. I would recommend resubmitting the manuscript with the data analysis.

2) Why are so many different assessment tools used? I added up the possibility of 168 question items or more. Is all of this necessary and relevant to the study's aims? Further rationale and explanation would be helpful, along with an estimate of the number of questions, and the length of time required to implement and respond to the assessments.

3) Why is Burmese the only language being used (rather than Karen, Karenni, Chin, etc.)? Does use of the Burmese language have a potential political subtext for minority groups who might have experienced oppression by the government? You could directly address the demographics within the migrant populations, and the role of language in this type of research.

Less significantly, please note slight differences in how the primary outcomes are described on p. 2 and p. 7. The primary outcomes could also be described in more detail, since it reads as though physical and emotional abuse are an intended primary outcome, rather than trying to prevent such behaviors.

This seems like an interesting study that could ultimately have useful results, with lessons for other contexts.

Thank you for your efforts to reduce child maltreatment.

7. PLOS authors have the option to publish the peer review history of their article (what does this mean?). If published, this will include your full peer review and any attached files.

Reviewer #1: No

Reviewer #2: No

---

## [Author Response · Author response to Decision Letter 0]

26 Sep 2023

Dear Dr. Zhou:

We thank you for your consideration of our manuscript and for the constructive and useful comments from reviewers. We are pleased to submit detailed responses to the reviewers’ comments and a revised manuscript for your consideration. 

In addition to the revisions made in response to reviewers’ comments, we have also made the following revisions to the protocol and study design since our original submission due to the evolving conditions in the study setting. These revisions and their rationale are outlined below:

1. Removal of adolescent reports at T2: The original protocol specified that adolescent children of adult participants will be surveyed at T2. Since submitting the original protocol, security and extreme weather conditions in the study setting have made field data collection more challenging. Upon consultation with the field data collection team and local partner organizations, we decided to remove adolescent data collection from the study design. Analysis of adolescent-reported data would have been exploratory in any case, given that we would only be able to collect data from a subsample of children who met the eligibility criteria of being between 12 and 17 years old. Hence, we would not have had statistical power to conduct a robust analysis using adolescent reported data. Given the challenges in the study setting and mindful of the wellbeing of the field team, we therefore made the decision to drop the adolescent survey. We will continue to conduct focus group discussions with adolescents, as described in the protocol, to obtain qualitative perspectives and experiences of adolescents.

2. Inclusion of parent-reported child internalizing and externalizing symptoms at T2: The original protocol specified that adolescent-reported internalizing and externalizing symptoms would be assessed at T2 as an exploratory outcome. Given the decision to drop the adolescent survey, we decided to add parent-reported child internalizing and externalizing symptoms at T2 as an exploratory outcome. Parent-reported child internalizing symptoms will be assessed using the Mood and Feelings Questionnaire and externalizing symptoms will be assessed using the externalizing subscale of the Child and Adolescent Behaviour Inventory (CABI). Further details can be found in the revised manuscript.

3. Timing of T2 data collection: Given the challenging conditions in the study setting, we decided to shift T2 data collection to be 4-months post intervention rather than 3-months post intervention. This change will allow the field teams more time to track and survey participants as well as ensure staff are able to maintain their well-being.

4. Other: Additional edits have been made to further clarify the Statistical Analysis section of the protocol. These edits do not substantively change the approach to analysis.

COMMENTS FROM REVIEWERS AND AUTHOR RESPONSES

Reviewer #1

This protocol for a randomized controlled trial addresses an important and understudied area of research—child maltreatment in migrant and displaced families, particularly in low- and middle-income countries (LMICs). It has the potential to fill a significant gap in the literature on the prevention of child maltreatment in vulnerable populations. The protocol clearly outlines the rationale, provides detailed information about the methods (including participants, measures, and procedures), and offers a precise plan for statistical analysis and discussion. The reasoning behind choosing a film-based intervention is clearly articulated, emphasizing its potential for cost-effective, large-scale implementation in resource-limited settings. The decision to use a pragmatic, superiority cluster randomized controlled trial will allow for the investigation of the intervention's effectiveness in the “real world.” I also appreciate the use of various measures (both parent reports and adolescent reports) to evaluate the primary and secondary outcomes of the intervention. Furthermore, the use of both quantitative and qualitative methods could offer valuable insights about intervention delivery, acceptability, perceived impacts, and potential mechanisms of change. Overall, I view the protocol very positively and I am looking forward to reading about the results of this important study.

I have only one minor comment:

How did the authors of the protocol ensure that the film is adequate for the needs and situations of families with children across such a wide age range, from 4 to 17 years?

Response to reviewer: Thank you for your review and feedback on the protocol. In response to your question, we worked closely with partner organization Sermpanya Foundation on the development of the script to ensure that the characters and parenting scenarios in the film addressed a wide age range and corresponding developmental stages. The film depicts two families, each with younger children (e.g. infant, 5-6 years) and older children (12-14 years). Scenes in the film depict different parent-child interactions that correspond to the child’s age and developmental stage. For example, scenes with younger children depict using play as way to foster learning, while scenes with older children focus on positive parent-adolescent communication to promote problem solving.

We have edited the manuscript as follows to provide further details (please see page 10):

“The characters and scenarios depicted in the film reflect different ages and developmental stages to ensure that the film would be relevant to families with children across a wide age range.”

Reviewer #2

The manuscript is well-written and addresses the important social issue of child maltreatment through a novel approach, and with a vulnerable population.

My questions and concerns about the manuscript are three-fold:

1) Why submit the article at this stage, before data has been collected and analyzed? While it appears to be a novel approach, the manuscript seems incomplete without an analysis of the results. I would recommend resubmitting the manuscript with the data analysis.

Response to reviewer: Thank you for your review and feedback on the protocol. In response to your question, we are submitting the protocol for publication in line with international commitments to improve research standards by promoting transparency, reducing publication bias, and enhancing the reproducibility of study design and analysis (see for example the UK Medical Research Council policy on open research data: clinical trials and public health intervention studies, October 2016). Upon completion of data collection and analysis, we will prepare and submit a separate manuscript reporting study results.

2) Why are so many different assessment tools used? I added up the possibility of 168 question items or more. Is all of this necessary and relevant to the study's aims? Further rationale and explanation would be helpful, along with an estimate of the number of questions, and the length of time required to implement and respond to the assessments.

Response to reviewer: Thank you for this question. The measures have been carefully selected to rigorously and robustly assess each of the primary, secondary, and exploratory outcomes theorized to be potentially impacted by the intervention. In addition, several measures (e.g. trauma exposure, daily stressors) are included to investigate potential moderating effects – for example, to examine if caregivers who have experienced more trauma benefit more or less from the intervention than caregivers who have experienced less trauma. There are approximately 100 items in the survey, which takes 1-2 hours to complete depending on the participants. This length is consistent with past surveys that the study team has conducted in this setting. We have added details about the survey length on page 15 of the revised manuscript. 

3) Why is Burmese the only language being used (rather than Karen, Karenni, Chin, etc.)? Does use of the Burmese language have a potential political subtext for minority groups who might have experienced oppression by the government? You could directly address the demographics within the migrant populations, and the role of language in this type of research.

Response to reviewer: Thank you for raising this very important point. We agree that conducting the study in Burmese only is a limitation and have now included this on page 25 of the revised manuscript. The decision to only use Burmese was due to practical considerations regarding time and resources. Unfortunately we did not have the ability to translate and administer the surveys in multiple languages. 

Less significantly, please note slight differences in how the primary outcomes are described on p. 2 and p. 7. The primary outcomes could also be described in more detail, since it reads as though physical and emotional abuse are an intended primary outcome, rather than trying to prevent such behaviors.

Response to reviewer: Thank you for this helpful suggestion. We have now edited the description of the primary outcomes to clarify further.

This seems like an interesting study that could ultimately have useful results, with lessons for other contexts.

Thank you for your efforts to reduce child maltreatment.

Response to reviewer: Thank you for taking the time to review the manuscript and for your thoughtful feedback.

Reviewer #3

This is an impressive protocol. It is very difficult to plan for an implement RCTs to address child maltreatment, and set this up in for addressing maltreatment in a highly vulnerable population is impressive. It will generate important data about a primary prevention strategy – using film-based intervention to improve positive parenting and reduce physical and emotional abuse is important and much needed. High income countries also lack similarly robust controlled studies, so it will be important globally in a number of ways.

Response to reviewer: Thank you for your review and feedback.

We hope these revisions are satisfactory and look forward to a favourable response.

---

## [Editor Report · Decision Letter 1]

17 Oct 2023

A film-based intervention to reduce child maltreatment among migrant and displaced families from Myanmar: Protocol of a pragmatic cluster randomized controlled trial

PONE-D-23-16654R1

Dear Dr. Amanda Sim,

We’re pleased to inform you that your manuscript has been judged scientifically suitable for publication and will be formally accepted for publication once it meets all outstanding technical requirements.

Kind regards,

Malgorzata Gambin

Guest Editor

PLOS ONE

Additional Editor Comments (optional):

I recently had the pleasure of reviewing your manuscript and I am pleased to inform you that I have been invited to act as a guest editor. I would like to thank you for the thoughtful and elaborate responses to my comments and those of the other reviewers. Your efforts have greatly contributed to the improvement of the manuscript. Congratulations! 

---

## [Editor Report · Acceptance letter]

20 Oct 2023

PONE-D-23-16654R1 

A film-based intervention to reduce child maltreatment among migrant and displaced families from Myanmar:
Protocol of a pragmatic cluster randomized controlled trial 

Dear Dr. Sim:

I'm pleased to inform you that your manuscript has been deemed suitable for publication in PLOS ONE. Congratulations! Your manuscript is now with our production department. 

Kind regards, 

on behalf of

Dr. Malgorzata Gambin 

Guest Editor

PLOS ONE